# A Hybrid 3D–2D Feature Hierarchy CNN with Focal Loss for Hyperspectral Image Classification

**Xiaoyan Wen [1], Xiaodong Yu [1,2,*], Yufan Wang [1], Cuiping Yang [1] and Yu Sun [3]**

1   School of Computer Science and Information Engineering, Harbin Normal University, Harbin 150025, China; keithmalarkey.lyx@gmail.com (X.W.); wangyufan1415@gmail.com (Y.W.); ycphsd@hrbnu.edu.cn (C.Y.)
2   Key Laboratory of Intelligent Education and Information Engineering, Heilongjiang Universities, Harbin 150025, China
3   Department of Municipal and Environmental Engineering, Heilongjiang Institute of Construction Technology, Harbin 150025, China; donglin_2016@nefu.edu.cn
*   Correspondence: yuxiaodong@hrbnu.edu.com; Tel.: +86-138-0458-6303

**Abstract:** Hyperspectral image (HSI) classification has been extensively applied for analyzing remotely sensed images. HSI data consist of multiple bands that provide abundant spatial information. Convolutional neural networks (CNNs) have emerged as powerful deep learning methods for processing visual data. In recent work, CNNs have shown impressive results in HSI classification. In this paper, we propose a hierarchical neural network architecture called feature extraction with hybrid spectral CNN (FE-HybridSN) to extract superior spectral–spatial features. FE-HybridSN effectively captures more spectral–spatial information while reducing computational complexity. Considering the prevalent issue of class imbalance in experimental datasets (IP, UP, SV) and real-world hyperspectral datasets, we apply the focal loss to mitigate these problems. The focal loss reconstructs the loss function and facilitates effective achievement of the aforementioned goals. We propose a framework (FEHN-FL) that combines FE-HybridSN and the focal loss for HSI classification and then conduct extensive HSI classification experiments using three remote sensing datasets: Indian Pines (IP), University of Pavia (UP), and Salinas Scene (SV). Using cross-entropy loss as a baseline, we assess the hyperspectral classification performance of various backbone networks and examine the influence of different spatial sizes on classification accuracy. After incorporating focal loss as our loss function, we not only compare the classification performance of the FE-HybridSN backbone network under different loss functions but also evaluate their convergence rates during training. The proposed classification framework demonstrates satisfactory performance compared to state-of-the-art end-to-end deep-learning-based methods, such as 2D-CNN, 3D-CNN, etc.

**Keywords:** hyperspectral remote sensing images; feature extraction; convolutional neural network (CNN); class imbalance; focal loss

## 1. Introduction

Hyperspectral images (HSIs) are generated by imaging spectrometers mounted on various space platforms, capturing spatial–spectral information [1]. The advancement of hyperspectral imaging technology has enabled sensors to capture hundreds of continuous spectral bands with nanometer-level resolution [2,3]. As a result, HSIs have found numerous applications in diverse fields, including environmental monitoring [4–6], hyperspectral anomaly detection [7], and hyperspectral image classification [8]. HSI classification, in particular, serves as a fundamental technique in many hyperspectral remote sensing applications, and it has proven to be invaluable in precision agriculture [9], geological exploration [10], and other domains [11,12].

The task of HSI classification is mainly tackled by two schemes: one with a hand-crafted feature extraction method and another with learning-based feature extraction method. In the early phase of HSI classification, the strategy of extracting more spectral

or spatial features is conventional machine learning. For instance, Yang and Qian [13] introduced a novel approach for hyperspectral image classification called multiscale joint collaborative representation with a locally adaptive dictionary. It constrains the adverse impact to HSI classification from useless pixels. Camps-Valls et al. [14] proposed and validated the effectiveness of composite kernels, a novel technique that combines multiple kernel functions to improve hyperspectral image classification performance. Other hand-crafted approaches, one after another, were proposed. One prominent method is the joint sparse model and discontinuity preserving relaxation [15]. The approach preprocesses each pixel and calculates relevant statistical measures, aiming at elegantly integrating spatial context and spectral signatures. Similarly, sparse self-representation [16] addresses band selection using optimization-based sparse self-representation, optimizing for efficient feature selection. To improve classification accuracy, researchers have proposed fusing correlation coefficient and sparse representation [17], aiming to harness the strengths of both methods. Additionally, multiscale superpixels and guided filter [18] have been explored for sparse-representation-based hyperspectral image classification, promising effective feature extraction and classification. The Boltzmann-entropy-based unsupervised band selection [19] has been investigated, targeting informative band selection to enhance classification performance.

However, the classification processes, based on the abovementioned approaches, are relatively cumbersome due to that their accomplishments rely on manually extracted features. Furthermore, faced with the inherent high-dimensional complexity of HSIs, the researchers find it hard to obtain ideal classification just using the above approaches, especially in challenging scenes [20]. Regrettably, the lack of labeled samples in the HSI field is in sharp contrast to the richness of spectral data. This fact poses the challenge of learning better feature representation and is prone to overfitting of methods. In view of the above problems, some schemes have been proposed to alleviate them, mainly including feature extraction [21–23], dimension reduction [24,25], and data augmentation [26].

Deep learning has emerged as a powerful method for feature extraction, enabling the identification of features from hyperspectral images (HSIs). Among various deep learning models, the convolutional neural network (CNN) stands out as one of the most widely applied models to address HSI classification challenges. CNNs have shown remarkable performance gains over conventional hand-designed features. CNN models are capable of processing spatial HSI patches as data inputs, leading to the development of progressive CNN-based methods that leverage both spectral and spatial features. For instance, Mei et al. [27] proposed a CNN model that adopts a pixel-wise approach and involves preprocessing each pixel by calculating the mean and standard deviation of the pixel neighborhood for each spectral band. On the other hand, Paoletti et al. [28] and Li et al. [29] introduced two distinct CNN models—one for extracting spatial features and the other for extracting spectral features. These models utilize a softmax classifier to achieve desirable classification results. While CNN-based technologies can effectively extract spatial and spectral features for HSI classification and other applications, they still encounter challenges in effectively utilizing information related to spatial and spectral associations.

In contrast, some technologies, which learn spatial and spectral features simultaneously for HSI classification, have been proposed. Yang et al. [30] proposed a multiscale wavelet 3D-CNN (MW-3D-CNN). Great importance was attached to the relationship information amid adjacent channels, while the corresponding model's calculating complexity was augmented intensively. Accordingly, Roy et al. [31] proposed the 3D–2D CNN feature hierarchy model for HSI classification. On one hand, a few 3D-CNN layers were utilized to extract spectral information amid spectral bands. On the other hand, 2D-CNN layers concentrated on much spatial texture and context. Liu et al. [32] extended the CNN model by incorporating the attention mechanism to enhance feature extraction from HSI. More recently, Zhong et al. [33] introduced the spectral–spatial residual network (SSRN). In the SSRN's designed residual blocks, identity mapping is utilized to connect 3D convolutional layers. These innovative models and technologies address the limitations of previous

approaches and effectively leverage spectral and spatial features to significantly improve the performance of HSI classification.

In recent times, large language models such as Transformer [34] have become a new paradigm in the field of natural language processing (NLP). Transformer introduces the attention mechanism, which allows for interactions between different tokens, capturing long-range semantic dependencies. Inspired by this success, researchers have extended these methods to the field of computer vision (CV). Similar to Transformer in NLP, Vision Transformer (ViT) [35] is typically pretrained on unlabeled image streams or video streams and then fine-tuned on downstream tasks to train model parameters. In the domain of remote sensing imagery, more researchers are adopting ViT or its variants [36] as foundational models for their studies. Liu et al. [37] utilized a customized Swin Transformer to reduce computational complexity and obtained strong generalization. Ayas and Tunc-Gormus [38] introduced a novel spectral Swin Transformer (SpectralSWIN) network, but without using attention mechanism, to hierarchically fuse spatial–spectral features and achieve a significant superiority. Zhao et al. [39] proposed a spectral–spatial axial aggregation Transformer framework to perform multiscale feature extraction and fusion on the input data while utilizing spectral shift operations to ensure information aggregation and feature extraction across different spectral components.

With regard to the issue of HSI classification, currently, many loss functions have been designed and obtained stunning performance. Multiclass hinge (MCH) loss in support vector machine (SVM), as one of the traditional classification strategies, is the main solution to HSI classification in the early phase. Wang et al. [40] introduced a novel classification framework, incorporating spatial, spectral, and hierarchical structure features, for hyperspectral images. The approach involves integrating three different and important types of information into the SVM classifier. By leveraging this joint integration, the proposed framework intends to enhance the classification performance of hyperspectral images. Furthermore, cross-entropy loss, as a standard metric, is widely applied to complete different classification tasks in hyperspectral image recognition, such as [41,42]. Regrettably, common and publicly available datasets Indian Pines (IP), Salinas (SA) and University of Pavia (UP), as well as the overwhelming majority of other hyperspectral imagery datasets, are class-imbalanced [43,44] (See Section 2.2). This fact poses great challenges for deep-learning-based HSI classification models regarding how to handle the class imbalance problem [45].

In this paper, spatial–spectral feature obtained by the deep learning approach is proposed for the HSI classification task. First, to extract more vital HSI features, including spatial and spectral information, we propose a novel convolutional neural network (FE-HybridSN), which pays attention to the correlation of adjacent bands from HSI. Compared with the plain 3D-CNN model for HSI classification, the FE-HybridSN alleviates the burden from complex computation. The proposed framework results in better HSI classification performance when the 3D-CNN model and the FE-HybridSN have similar scale in model structure. Second, to cope with the class-imbalanced problem in HSI classification, we apply the focal loss as loss function. In fact, we obtain expected classification results based on the focal loss after much rigorous research. In summary, the main contributions of this paper are as follows:

1.  We fashion a novel five-layer FE-HybridSN of hyperspectral images for mining spatial context and spectral features;
2.  We apply the focal loss as the loss function to alleviate the class-imbalanced problem in the HSI classification task;
3.  We explore feature learning and classification of hyperspectral images using systematic experiments, and inspire new deep learning ideas for hyperspectral applications.

The paper is structured as follows. Section 2 outlines the challenges and presents our proposed approach to address them. In Section 3, we provide a detailed description of the experimental background and present the fundamental experimental configurations. Additionally, we compare our method with other state-of-the-art approaches. Building

upon the results presented in Section 3, Section 4 offers a comprehensive discussion to provide insights and interpretations. Finally, in Section 5, we summarize the findings of the paper and propose promising research directions for future exploration.

## 2. Methodology

We present a novel joint network called FE-HybridSN, which combines a hybrid 3D–2D CNN architecture with the focal loss to achieve hyperspectral image classification. The proposed approach is designed to capitalize on spectral–spatial feature maps and extract more abstract representations from hierarchical space, as depicted in Figure 1. By leveraging the hybrid 3D–2D CNN model, we aim to effectively mine valuable information from the hyperspectral data. Additionally, we utilize the focal loss to mitigate any adverse effects on hyperspectral image classification, as illustrated in Figure 1. The focal loss helps in handling challenging samples, thereby enhancing the overall performance of the classification process.

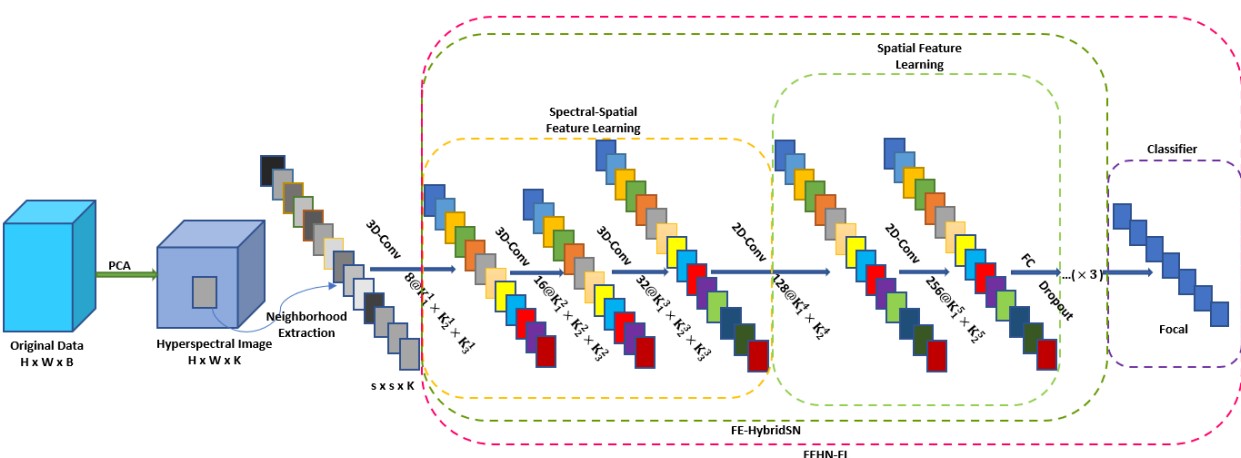

**Figure 1.** Overview of the proposed feature extractor FE-HybridSN and entire framework FEHN-FL. Note that batch normalization (BN) and rectified linear unit (ReLU), following every convolutional operation, are omitted in the figure.

### 2.1. Proposed Model

Assume the spatial–spectral hyperspectral data cube is denoted by $I_{origin} \in \mathbb{R}^{H \times W \times B}$, where $H$, $W$, and $B$ represent the height, width, and the number of spectral bands in the hyperspectral image (HSI), respectively. Each pixel in $I_{origin}$ is associated with a one-hot label vector $Y = \{y_1, y_2, ..., y_C\} \in \mathbb{R}^{1 \times 1 \times C}$, where $C$ signifies the number of land cover classes. However, a real-world challenge arises from the fact that high-dimensional hyperspectral pixels may exhibit mixtures of multiple land cover classes, leading to substantial spectral intravariability and significant interclass similarity.

To address this issue, we employ principal component analysis (PCA) to preprocess the original HSI data $I_{origin}$ and eliminate redundant spectral information. PCA reduces the original $B$ spectral bands to $K$ bands while preserving the spatial dimensions, thus retaining the spatial information even after the PCA process. Consequently, we represent the processed spatial–spectral hyperspectral data cube as $I_{PCA} \in \mathbb{R}^{H \times W \times K}$. Here, $H$ and $W$ still represent the height and width of the spectral bands, and $K$ indicates the number of retained spectral bands after PCA processing.

During the data preprocessing stage, besides reducing the dimensionality of the hyperspectral data using PCA, it is also necessary to segment the image into small, overlapping 3D blocks. The primary purpose of this is to apply our deep learning method on each smaller 3D block. Each adjacent 3D block originates from the PCA-reduced hyperspectral data cube, and they are uniquely identified by the central spatial coordinates. We denote each 3D block as $I_{block} \in \mathbb{R}^{s \times s \times K}$, where $s$ represents the size of each spatial window and $K$

denotes the depth of spectral bands in each block. Notably, $K$ also corresponds to the number of bands retained after PCA dimensionality reduction. Ultimately, the PCA-reduced hyperspectral data cube generates $(H - s + 1) \times (W - s + 1)$ 3D blocks. Specifically, for any given hyperspectral 3D block with central spatial coordinates $(\alpha, \beta)$, the corresponding spatial window will cover the width range $[\alpha - (s - 1)/2, \alpha + (s - 1)/2]$ and the height range $[\beta - (s - 1)/2, \beta + (s - 1)/2]$.

We propose a framework named FEHN-FL with hierarchical convolutional structure for HSI classification. CNN parameters are trained using supervised methods [46] with gradient descent optimization. Conventional 2D CNNs compute 2D discriminative feature maps by applying convolutions solely over spatial dimensions, but they lack the ability to identify and handle spectral information. In contrast, a single 3D-CNN can slide the convolutional kernel along all three dimensions (height, width, and spectral) and interact with both spatial and spectral dimensions. This capability allows the 3D-CNN to comprehensively capture spatial and spectral information from the high-dimensional hyperspectral data. FE-HybridSN, the backbone of the FEHN-FL, hierarchically combines three 3D convolutions (refer to Equation (2)), two 2D convolutions (refer to Equation (1)), and three fully connected layers, achieving a balanced integration of spectral and spatial information for more effective HSI classification.

In 2D-CNN, the output results are generated through the process of convolution, where the input image is convolved with 2D filters such that the size is predesigned, also known as convolutional kernels. This operation involves element-wise multiplication between the filter's weights and the pixels in the input image, followed by summation to obtain the new output pixel value. The convolutional process slides the filter across the entire input image, computing the output pixel value at each position. By utilizing distinct filters at different layers, 2D-CNN can effectively learn various features present in the image. These learned features are then combined to form higher-level representations, enabling the model to perform image classification and feature extraction tasks. The resulting features from the convolution are passed through an activation function, introducing nonlinearity to the model. In 2D convolution, the activation value of the $j$th feature map at spatial position $(x, y)$ in the $i$th layer is represented as $v_{(i,j)}^{(x,y)}$ and can be expressed by the following equation:

$$v_{(i,j)}^{(x,y)} = \psi \left( b_{i,j} + \sum_{\tau=1}^{d_{l-1}} \sum_{\rho=-\gamma}^{\gamma} \sum_{\sigma=-\delta}^{\delta} w_{i,j,\tau}^{\sigma,\rho} \times v_{i-1,\tau}^{x+\sigma,y+\rho} \right) \tag{1}$$

where $\psi$ is the nonlinear activation function, $b_{i,j}$ is the bias parameter for the $j$th feature map of the $i$th layer, and $d_\tau$ indicates the number of feature maps in the $\tau$th layer. The size of the predesigned convolutional kernel is $(2\gamma + 1) \times (2\delta + 1)$. $w_{i,j}$ corresponds to the weight parameter for the $j$th feature map of the $i$th layer.

According to the definition of three dimensional convolution [47], we perform convolutional operations by applying 3D convolutional kernels to the hyperspectral images. In the FEHN-FL, the feature maps in the convolutional layer are generated by applying 3D convolutional kernels on discrete or consecutive spectral bands of the input layer, thus capturing spectral information. In 3D convolution, the activation value $v_{(i,j)}^{(x,y,z)}$ of the $j$th feature map at spatial position $(x, y, z)$ in the $i$th layer is expressed as follows:

$$v_{(i,j)}^{(x,y,z)} = \psi \left( b_{i,j} + \sum_{\tau=1}^{d_{l-1}} \sum_{\lambda=-\eta}^{\eta} \sum_{\rho=-\gamma}^{\gamma} \sum_{\sigma=-\delta}^{\delta} w_{i,j,\tau}^{\sigma,\rho,\lambda} \times v_{i-1,\tau}^{x+\sigma,y+\rho,z+\lambda} \right) \tag{2}$$

where $2\eta + 1$ is the depth of kernel along the spectral dimension and other parameters are the same as in Equation (1).

In the FE-HybridSN, we design different 3D convolution kernels. We denote the structure of the 3D kernel as $L@K_1^u \times K_2^u \times K_3^u$, where u and L, respectively, represent the layer index of current kernels and the number of output channels for the current convolution layer. Moreover, we denote the structure of the 2D kernel as $L@K_1^d \times K_2^d$,

where d represents the layer index of current kernels, and L is same as the 3D kernel. One of our designed kernel sequence is (8@3 × 3 × 7, 16@3 × 3 × 5, 32@3 × 3 × 3, 128@3 × 3, 256@3 × 3). Here, 8@3 × 3 × 7 means that the first 3D kernel size is 3 × 3 × 7 and the number of output channel is eight when a single channel from *P* is the first input. The explanation of subsequent kernel parameters is the same as 8@3 × 3 × 7. The proposed model is comprehensively summarized in Table 1, presenting the layer index, output map dimensions, and the corresponding number of parameters.

**Table 1.** Lay-wise summary of the proposed FE-hybrid architecture with spatial size 15 × 15.

| Layer-Index | Output Shape | # Parameter |
| --- | --- | --- |
| Input-0 | (1, 15, 15, 15) | - |
| Conv3d-1 | (8, 9, 13, 13) | 512 |
| BatchNorm3d-2 | (8, 9, 13, 13) | 16 |
| ReLU-3 | (8, 9, 13, 13) | 0 |
| Conv3d-4 | (16, 5, 11, 11) | 5776 |
| BatchNorm3d-5 | (16, 5, 11, 11) | 32 |
| ReLU-6 | (16, 5, 11, 11) | 0 |
| Conv3d-7 | (32, 3, 9, 9) | 13,856 |
| BatchNorm3d-8 | (32, 3, 9, 9) | 64 |
| ReLU-9 | (16, 5, 11, 11) | 0 |
| Conv2d-10 | (128, 7, 7) | 110,720 |
| BatchNorm2d-11 | (128, 7, 7) | 256 |
| ReLU-12 | (128, 7, 7) | 0 |
| Conv2d-13 | (256, 5, 5) | 295,168 |
| BatchNorm2d-14 | (256, 5, 5) | 512 |
| ReLU-15 | (256, 5, 5) | 0 |
| Linear-16 | (256) | 1,638,656 |
| Dropout-17 | (256) | 0 |
| Linear-18 | (128) | 32,896 |
| Dropout-19 | (128) | 0 |
| Linear-20 | (16) | 2064 |

Total parameters: 2,100,528
Trainable parameters:
2,100,528
Nontrainable parameters: 0

The parameters are based on the Indian Pines (IP) dataset.

In the proposed method, the batch normalization (BN) layers are introduced as the important elements since they make the distribution of input data in every layer of the network relatively stable and accelerate the training speed of the entire framework. The formula of BN is as follows:

$$N(x) = \frac{x - \bar{x}}{\sqrt{D(x) + \varepsilon}} \cdot \gamma + \beta \tag{3}$$

where $\bar{x}$ is the average of the summation, $D(x)$ is the variance, $\gamma$ and $\beta$ are learnable parameter vectors, and $\varepsilon$ is a parameter for numerical stability. In addition, the nonlinear layer aims at adding some nonlinear features to the network. Then, it is notable that the rectified linear unit (ReLU) [48] is selected into each 3D/2D convolutional layer.

### 2.2. Focal Loss

The class-imbalanced problem commonly occurs in tasks with a long-tailed data distribution, where a few classes dominate the data, while most classes have very few samples. In traditional classification and visual recognition tasks, the training distribution can be balanced through manual intervention using resampling strategies. This strategy ensures that the number of samples from different classes does not significantly differ. However, as

the number of categories increases, maintaining a balance between all categories becomes increasingly challenging, resulting in an exponential growth in the collection of samples.

In the case of HSI classification, neither using resampling strategies nor not using them are feasible or rational solutions. Hence, we employ the focal loss as the loss function. Compared to traditional loss functions like multiclass hinge loss and cross-entropy loss, the focal loss provides better performance and tackles the class imbalance issue effectively.

### 2.2.1. Balanced Cross Entropy

Introducing weighted factors $\alpha$ and $1 - \alpha$ for positive and negative classes, respectively, is a common approach to address class imbalance issues. In practice, $\alpha$ is often initialized as the inverse class frequency, which is the reciprocal of the ratio of positive class samples to negative class samples. The purpose is to assign larger weights to the classes with fewer samples during the training phase, aiming to balance the class distribution. We denote the $\alpha$-balanced cross entropy (CE) loss as

$$\text{CE}(p, y) = \text{CE}(p_t) = -\alpha_t \log(p_t). \tag{4}$$

where $p_t$ is expressed as follows:

$$p_t = \begin{cases} p, & y = 1 \\ 1 - p, & y = 0 \end{cases} \tag{5}$$

Here, the definition of $\alpha_t$ is analogous to how we defined $p_t$, and $y \in \{0, 1\}$ indicates negative and positive class, respectively.

### 2.2.2. Focal Loss Definition

Easily classified negatives comprise the majority of the loss and dominate the gradient [49]. While $\alpha$ balances the importance of positive/negative examples, it does not differentiate between easy and hard examples. The focal loss reconstructs the balanced cross-entropy loss to down-weight easy examples and thus focuses training on hard negatives. The focal loss is defined as follows:

$$\text{FL}(p_t) = -(1 - p_t)^\gamma \log(p_t). \tag{6}$$

Formally, we use an $\alpha$-balanced variant of the focal loss as the loss function:

$$\text{FL}(p_t) = -\alpha_t(1 - p_t)^\gamma \log(p_t). \tag{7}$$

## 3. Experiments

### 3.1. Description of Experiment Datasets

Extensive experiments on three benchmark datasets of HSIs, including Indian Pines (IP) [50], University of Pavia (UP) [51], and Salinas Valley (SV) [52], were conducted to assess the classification performance of the proposed framework. Table 2 provides a concise overview of the hyperspectral image (HSI) datasets that were considered in the study. It includes information such as the number of samples available per class in each dataset and the corresponding ground truth (GT) provided.

### 3.2. Experimental Configuration

The proposed framework (FEHN-FL) was compared to other methods available in the literature. Concretely, we compared the backbone FE-HybridSN used for feature extraction with the following methods: (1) 2D-CNN [53]; (2) 3D-CNN [54]; (3) M3D-CNN [55]; (4) 3D2D-HybridSN [31]. In addition, we introduced other loss functions as experimental comparison objects, including cross-entropy (CE) loss and multiclass hinge (MCH) loss in support vector machine (SVM), to evaluate the performance of the focal loss. For each method, we carefully fine-tuned and fixed all hyperparameters to their optimal

values before conducting the experiments. This ensured that we obtained the best possible performance from each approach to facilitate a fair comparison of the results. The details are shown in Table 3.

**Table 2.** The total number of samples and the sample quantity for each land cover.

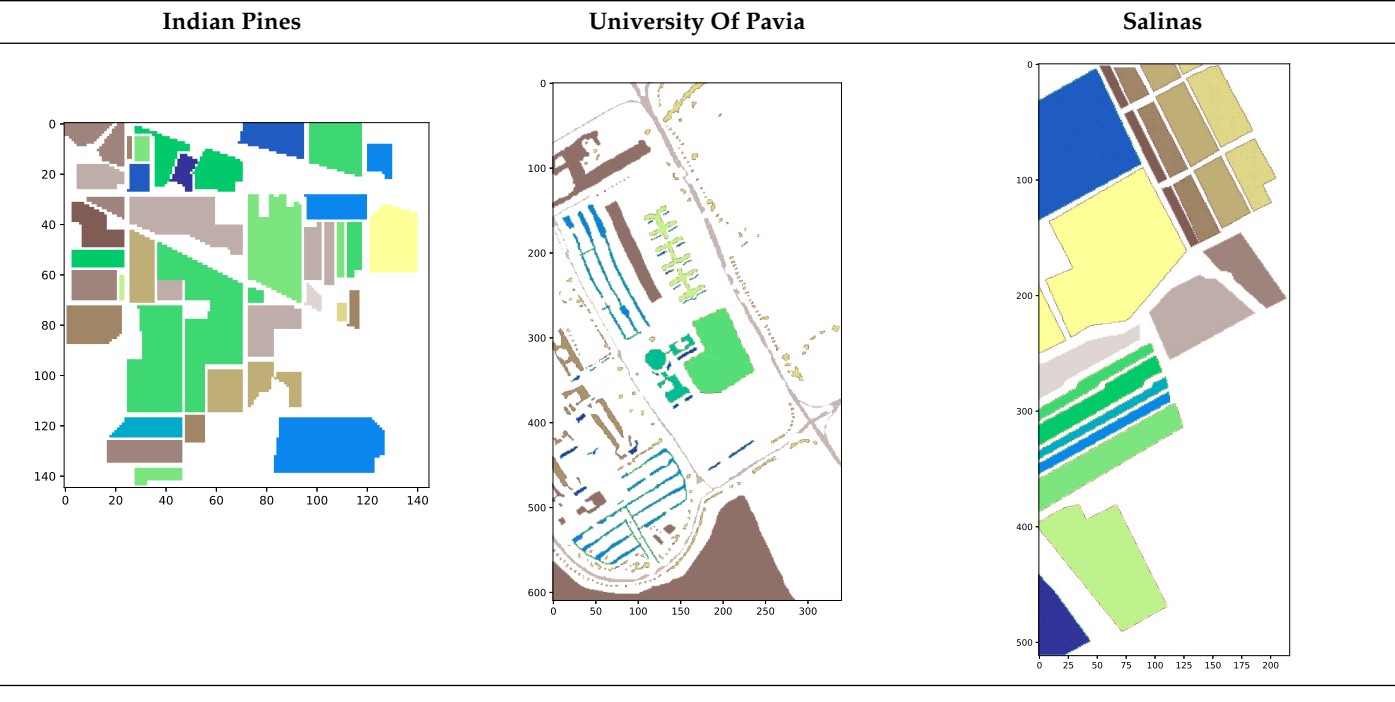

| Color | Land-Cover type | Samples | Color | Land-Cover type | Samples | Color | Land-Cover type | Samples |
|---|---|---|---|---|---|---|---|---|
| | Background | 10,776 | | Background | 164,624 | | Background | 56,975 |
| | Alfalfa | 46 | | Asphalt | 6631 | | Brocoli-green-weeds-1 | 2009 |
| | Corn-notill | 1428 | | Meadows | 18,649 | | Brocoli-green-weeds-2 | 3726 |
| | Corn-min | 830 | | Gravel | 2099 | | Fallow | 1976 |
| | Corn | 237 | | Trees | 3064 | | Fallow-rough-plow | 1394 |
| | Pasture | 483 | | Painted metal sheets | 1345 | | Fallow-smooth | 2678 |
| | Trees | 730 | | Bare Soil | 5029 | | Stubble | 3959 |
| | Pasture-mowed | 28 | | Bitumen | 1330 | | Celery | 3579 |
| | Hay-windrowed | 478 | | Self-Blocking Bricks | 3682 | | Grapes-untrained | 11,271 |
| | Oats | 20 | | Shadows | 947 | | Soil-vineyard-develop | 6203 |
| | Soybean-notill | 972 | | | | | Corn-senesced-green-weeds | 3278 |
| | Soybean-min | 2455 | | | | | Lettuce-romaine-4wk | 1068 |
| | Soybean-clean | 593 | | | | | Lettuce-romaine-5wk | 1927 |
| | Wheat | 205 | | | | | Lettuce-romaine-6wk | 916 |
| | Woods | 1265 | | | | | Lettuce-romaine-7wk | 1070 |
| | Bldgs-Grass-Trees-Drives | 386 | | | | | Vinyard-untrained | 7268 |
| | Stone-Steel towers | 93 | | | | | Vinyard-vertical-trellis | 1807 |
| | **Total samples** | **21,025** | | **Total samples** | **207,400** | | **Total samples** | **111,104** |

**Table 3.** Training hyperparameter settings in the FEHN-FL.

| | IP | UP | SV |
|---|---|---|---|
| Epochs | | 150 | |
| Optimizer | | Adam [56] | |
| Batch size | 53 | 128 | 128 |
| Learning rate (LR) | | $1 \times 10^{-3}$ | |
| $\alpha_t$ and $\gamma$ | | 1 and $1 \times 10^{-6}$ | |

Before original data were used the input of framework FEHN-FL, PCA was applied over the hyperspectral data so as to reduce the dimension of the original hyperspectral image and extract principal components. In our experiments, all principal components were set as fifteen. According to the experimental design, the datasets IP, UP, and SV were randomly divided into two parts in proportion: 5%, 10%, and 15% of IP, UP, or SV labeled samples were the training samples, and the remainder were testing datasets. Considering different schemes about backbone and loss function, as well as the aforementioned datasets, we implemented sets of experiments to validate the performance of the FE-HybridSN based on CE loss (Experiment 1). Then, to test the effects rooted from different patch sizes, we correspondingly designed four different spatial sizes for input data (Experiment 2). Furthermore, we considered different training data percentages (Experiment 3) aimed at comparing the FEHN-FL to other state-of-the-art methods. Finally, we compared the focal loss with other loss functions and studied its convergence speed.

1.  In the first experiment, the FE-HybridSN was compared with 2D-CNN [53], 3D-CNN [54], M3D-CNN [55], and 3D2D-HybridSN [31] classification methods using a training set accounting for 15% of the whole labeled dataset over IP, UP, and SV datasets. Additionally, we set the input spatial size to be $15 \times N \times N$ (N could be 11, 15, 19, or 23) for the 2D-CNN [53], 3D-CNN [54], M3D-CNN [55], 3D2D-HybridSN [31], and FE-HybridSN, with N being the spatial size (i.e., patch size).

2.  In the second experiment, the FEHN-FL was compared with the four aforementioned methods. We designed four different patch sizes, i.e., $11 \times 11, 15 \times 15, 19 \times 19$, and $23 \times 23$. Here, we considered 10% of available labeled data for the IP, UP, and SV datasets.

3.  In the third experiment, we compared the proposed framework (FEHN-FL) with all the aforementioned methods [31,53–55]. The configurations of patch size were the same as the second experiment. We still focused on the performance just using our backbone (FE-HybridSN) with CE loss.

4.  In the final experiment, we compared the convergence speed of different loss functions, including cross-entropy (CE) loss, multiclass hinge (MCH) loss, and focal loss, using training samples of the same scale on the Indian Pines (IP), University of Pavia (UP), and Salinas Valley (SV) datasets.

In our experimental evaluation, we measured the performance using three metrics: OA, AA, and kappa. OA represents the ratio of correctly predicted samples to the total testing samples, providing an overall accuracy measure. AA computes the average classification accuracy across different classes, giving insight into the performance of individual classes. Kappa serves as a statistical measure, indicating the agreement between the ground truth map and the classification map. The experiments were conducted on a hardware setup comprising an Intel Core processor (2.30 GHz), 16 GB of memory, and an NVIDIA GeForce GTX 1050Ti graphics processing unit with 8 GB RAM. To implement the HSI classification methods, we utilized Python in the Pytorch platform; specifically, Pytorch 1.12.0, CUDA 11.7, cuDNN 8.0, and Python 3.8 were the specific research environments used.

### 3.3. Experimental Results

#### 3.3.1. Experiment 1

Table 4 presents the classification results with 15% training samples for the IP dataset. Specifically, the first column of Table 4 indicates other methods compared with the FE-HybridSN. The next four columns show the results of the OA, AA, and kappa coefficient, which were derived from using different spatial sizes: $11 \times 11, 15 \times 15, 19 \times 19$, and $23 \times 23$. In addition, Table 5 presents the classification results with 5% training samples for the UP dataset. The explanation for every item in Table 5 is the same as Table 4. Significantly, the last row contains the accuracies of the proposed backbone (i.e., FE-HybridSN).

**Table 4.** Classification accuracies (%) by different methods for IP dataset with 15% labeled training samples and various spatial sizes. Note that the bold represents the best performance with corresponding experimental configuration for different methods.

| Methods | 11 × 11 | | | 15 × 15 | | | 19 × 19 | | | 23 × 23 | | |
|---|---|---|---|---|---|---|---|---|---|---|---|---|
| | OA | AA | Kappa | OA | AA | Kappa | OA | AA | Kappa | OA | AA | Kappa |
| 2D-CNN [53] | 88.38 | 84.82 | 86.7 | 98.28 | 98.25 | 98.03 | 93.42 | 90.34 | 92.48 | 95.98 | 95.85 | 95.41 |
| 3D-CNN [54] | 96.75 | 96.18 | 96.29 | 98.82 | 98.96 | 98.65 | 98.55 | 98.25 | 98.35 | 98.24 | 98.42 | 98.00 |
| 3D2D-HybridSN [31] | 97.23 | 97.01 | 96.84 | 98.55 | 98.56 | 98.39 | 98.60 | 98.52 | 98.40 | 98.21 | 98.41 | 98.30 |
| M3D-CNN [55] | 96.90 | 95.29 | 96.46 | 98.07 | 96.63 | 97.80 | 98.30 | 97.97 | 98.06 | 98.35 | 95.54 | 98.12 |
| FE-HybridSN | **97.41** | **97.85** | **97.04** | **98.85** | **99.02** | **98.71** | **98.65** | **98.56** | **98.46** | **98.77** | **99.11** | **98.60** |

**Table 5.** Classification accuracies (%) by different methods for UP dataset with 5% labeled training samples and various spatial sizes. Note that the bold represents the best performance with corresponding experimental configuration for different methods.

| Methods | 11 × 11 | | | 15 × 15 | | | 19 × 19 | | | 23 × 23 | | |
|---|---|---|---|---|---|---|---|---|---|---|---|---|
| | OA | AA | Kappa | OA | AA | Kappa | OA | AA | Kappa | OA | AA | Kappa |
| 2D-CNN [53] | 95.71 | 93.75 | 94.32 | 99.15 | 98.79 | 98.87 | 98.36 | 97.23 | 97.83 | 95.80 | 94.08 | 94.45 |
| 3D-CNN [54] | 98.99 | 98.43 | 98.66 | 99.53 | 99.28 | 99.38 | 99.53 | 99.19 | 99.38 | 99.49 | 98.90 | 99.32 |
| 3D2D-HybridSN [31] | 99.40 | 99.02 | 99.20 | 99.38 | 99.20 | 99.17 | 99.59 | 99.37 | 99.46 | 99.70 | 99.37 | 99.60 |
| M3D-CNN [55] | 99.08 | 98.17 | 98.78 | 99.51 | 99.15 | 99.46 | 99.57 | 99.18 | 99.44 | 99.65 | 99.13 | 99.54 |
| FE-HybridSN | **99.47** | **99.13** | **99.30** | **99.61** | **99.40** | **99.48** | **99.73** | **99.50** | **99.64** | **99.73** | **99.37** | **99.64** |

Furthermore, as shown in Figure 2, we conducted two sets of experiments on the SV dataset based on spatial size. Two abscissae and two ordinates, respectively, indicate the names of different methods and classification accuracies (in percentages). Under the condition using established methods, we compared the classification performance using two different spatial sizes (11 × 11 and 23 × 23) and 10% training samples. The left one represents the classification result using the spatial size 11 × 11. Correspondingly, the right one represents the classification result using the spatial size 23 × 23.

### 3.3.2. Experiment 2

In the second experiment, we considered 5%, 10%, and 15% of the labeled data over the SV and UP datasets as training samples, and compared the classification performance of the proposed method with those of 2D-CNN [53], 3D-CNN [54], and 3D2D-HybridSN [31]. Table 6 shows the classification accuracies with 5%, 10%, and 15% labeled training samples and 11 × 11 spatial size by the abovementioned scheme.

### 3.3.3. Experiment 3

In order to comprehensively consider the performance of feature extractors and loss functions for hyperspectral image classification, we compared the proposed framework (FEHN-FL) with other methods. Concretely, spatial size 23 × 23 and three different training ratios (5%, 10%, and 15%) were considered for the SV and UP datasets. Table 7 presents the classification results according to the aforementioned experimental scheme.

### 3.3.4. Experiment 4

In Table 8, we propose the trainable parameter scales of various backbones. Table 9 demonstrates that when utilizing 10% labeled samples for training and setting the spatial size to 23 × 23, our proposed framework with the feature extractor FE-HybridSN achieves the minimum training time overhead.

**Table 6.** Classification accuracies (%) by different methods over SV and UP datasets with $11 \times 11$ spatial size and different training ratios. Note that the bold represents the best performance with corresponding experimental configuration for different methods.

| Training Ratio | Methods | SV | | | UP | | |
|---|---|---|---|---|---|---|---|
| | | OA | AA | Kappa | OA | AA | Kappa |
| 15% | 2D-CNN [53] | 98.91 | 99.42 | 98.78 | 98.43 | 97.48 | 97.91 |
| | 3D-CNN [54] | 99.90 | 99.88 | 99.89 | 99.69 | 99.44 | 99.59 |
| | 3D2D-HybridSN [31] | 99.90 | 99.88 | 99.89 | 99.32 | 99.09 | 99.09 |
| | FE-HybridSN | **99.91** | **99.89** | **99.90** | **99.73** | **99.66** | **99.64** |
| 10% | 2D-CNN [53] | 98.79 | 99.39 | 98.65 | 97.21 | 96.42 | 96.30 |
| | 3D-CNN [54] | 99.77 | 99.87 | 99.74 | 99.34 | 99.06 | 99.12 |
| | 3D2D-HybridSN [31] | 99.82 | 99.85 | 99.80 | 99.71 | 99.60 | 99.62 |
| | FE-HybridSN | **99.85** | **99.92** | **99.83** | **99.84** | **99.77** | **99.79** |
| 5% | 2D-CNN [53] | 98.21 | 99.02 | 98.01 | 96.05 | 95.05 | 94.77 |
| | 3D-CNN [54] | 98.92 | 99.18 | 98.80 | 99.03 | 98.48 | 98.71 |
| | 3D2D-HybridSN [31] | 98.48 | 98.46 | 98.31 | 99.21 | 98.97 | 98.95 |
| | FE-HybridSN | **98.92** | **99.22** | **99.45** | **99.45** | **99.17** | **99.27** |

**Table 7.** Classification accuracies (%) by different methods combined with CE, MCH, and focal loss over SV and UP datasets with $23 \times 23$ spatial size and different training ratios. Note that the bold represents the best performance with corresponding experimental configuration for different methods.

| Training Ratio | Methods | SV | | | UP | | |
|---|---|---|---|---|---|---|---|
| | | OA | AA | Kappa | OA | AA | Kappa |
| 15% | 2D-CNN [53] + CE | 99.95 | 99.89 | 99.94 | 99.09 | 98.71 | 98.80 |
| | 3D-CNN [54] + CE | 99.99 | 99.99 | 99.99 | 99.91 | 99.84 | 99.88 |
| | 3D2D-HybridSN [31] + CE | 99.99 | 99.99 | 99.99 | 99.95 | 99.91 | 99.94 |
| | M3D-CNN [55] + CE | 99.99 | 99.99 | 99.99 | 99.87 | 99.79 | 99.82 |
| | FE-HybridSN + CE | 99.97 | 99.95 | 99.98 | 99.93 | 99.91 | 99.91 |
| | FE-HybridSN + MCH | 99.96 | 99.93 | 99.96 | 99.24 | 99.01 | 98.99 |
| | Proposed | **99.99** | **99.99** | **99.99** | **99.95** | **99.95** | **99.94** |
| 10% | 2D-CNN [53] + CE | 99.89 | 99.89 | 99.88 | 98.14 | 97.12 | 97.54 |
| | 3D-CNN [54] + CE | 99.94 | 99.94 | 99.94 | 99.74 | 99.21 | 99.65 |
| | 3D2D-HybridSN [31] + CE | 99.94 | 99.93 | 99.93 | 99.89 | 99.77 | 99.84 |
| | M3D-CNN [55] + CE | 99.93 | 99.90 | 99.92 | 99.83 | 99.54 | 99.77 |
| | FE-HybridSN + CE | 99.96 | 99.94 | 99.93 | 99.89 | 99.79 | 99.83 |
| | FE-HybridSN + MCH | 99.85 | 99.94 | 99.84 | 99.72 | 99.53 | 99.64 |
| | Proposed | **99.96** | **99.95** | **99.96** | **99.89** | **99.79** | **99.85** |
| 5% | 2D-CNN [53] + CE | 98.86 | 99.17 | 98.85 | 95.80 | 94.08 | 94.45 |
| | 3D-CNN [54] + CE | 99.68 | 99.81 | 99.75 | 99.49 | 98.90 | 99.32 |
| | 3D2D-HybridSN [31] + CE | 99.64 | 99.79 | 99.60 | 99.70 | 99.37 | 99.60 |
| | M3D-CNN [55] + CE | 99.60 | 99.03 | 99.47 | 99.65 | 99.13 | 99.54 |
| | FE-HybridSN + CE | 99.34 | 99.67 | 99.27 | 99.73 | 99.37 | 99.64 |
| | FE-HybridSN + MCH | 99.55 | 99.75 | 99.50 | 99.36 | 98.61 | 99.15 |
| | Proposed | **99.75** | **99.88** | **99.79** | **99.75** | **99.43** | **99.65** |

**Table 8.** Trainable parameter scales for different backbone networks.

| | 2D-CNN [53] | 3D-CNN [54] | M3D-CNN [55] | 3D2D-HybridSN [31] | Proposed |
|---|---|---|---|---|---|
| Params | 0.19 M | 0.51 M | 0.99 M | 1.79 M | 2.19 M |

**Table 9.** The average training time using 10% labeled training samples with 23 × 23 spatial size over IP, UP, and SV datasets. Note that the minimum average training time is highlighted in bold.

|  | Focal Loss (min) | CE Loss (min) | MCH (min) |
|---|---|---|---|
| IP | **4.12** | 4.13 | 4.22 |
| UP | **14.93** | 15.35 | 15.01 |
| SV | **18.82** | 21.77 | 19.08 |

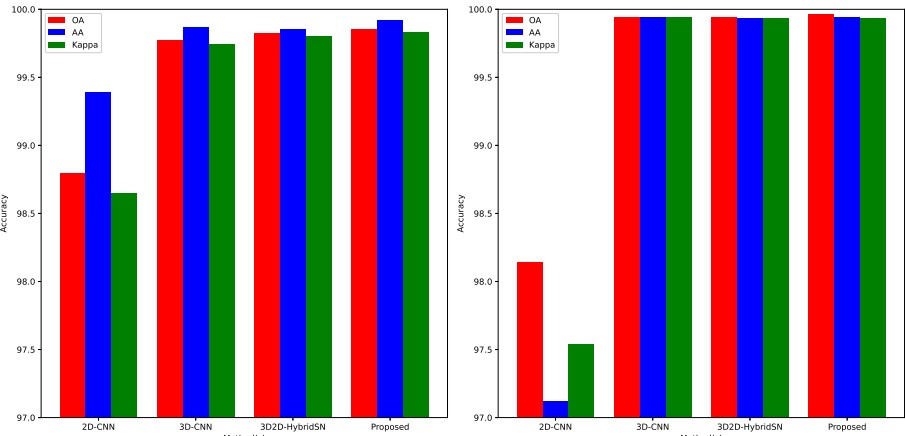

**Figure 2.** Classification accuracies (%) by different methods with 10% training samples and different spatial sizes over SV dataset (the left one is 11 × 11, another one is 23 × 23).

## 4. Discussion

After analyzing the results of Experiment 1 (refer to Section 3.3.1), several key observations can be made. Firstly, the FE-HybridSN, combined with cross-entropy loss, demonstrates high classification accuracy. When training with 15% and 5% labeled samples using different spatial sizes on the corresponding IP and UP datasets, our method outperforms other state-of-the-art techniques in terms of classification accuracy. Specifically, utilizing the CE loss as the terminal loss function and following the basic experimental configurations, the FE-HybridSN consistently achieves superior pixel-level classification results compared to 2D-CNN [53], 3D-CNN [54], 3D2D-HybridSN [31], and M3D-CNN [55].

In addition, Tables 4 and 5 highlight the significant impact of spatial size on HSI classification accuracy. Notably, the classification accuracies obtained using 19 × 19 and 23 × 23 patch sizes are consistently lower than those achieved with a 15 × 15 patch size across all methods in Table 4. Similarly, Table 5 illustrates that increasing the patch size leads to improved classification accuracy for the FE-HybridSN framework utilizing cross-entropy loss on the UP dataset. This finding emphasizes the crucial role of spatial size in HSI classification and its ability to adjust the decision boundary. Specifically, a smaller spatial size leads to a greater loss of information, resulting in lower classification accuracy, as evidenced by the experimental results. Furthermore, Figure 2 provides a visual representation of the relationship between spatial size and classification accuracy, further supporting these conclusions.

Based on the results in Table 6 of Experiment 2 (refer to Section 3.3.2), it is evident that our proposed feature extractor, FE-HybridSN, achieves significantly better classification accuracy on both the SV and UP datasets. On one hand, when using training samples with different proportions of labeled data, it is evident that as the number of training samples decreases, the classification accuracy of hyperspectral images also decreases. With the small scale of the training set, the risk of overfitting increases. In other words, when training is completed, the resulting classification decision boundaries may lack robust generalization ability, particularly when facing new samples such as test samples. On the other hand, with the same training ratio, the OA of the SV dataset is noticeably higher compared to

the classification accuracy of the UP dataset. This is primarily due to the fact that the SV dataset has a significantly larger training scale than the UP dataset.

In Experiment 3 (refer to Section 3.3.3), we considered the use of the focal loss as the terminal loss function. As depicted in Table 7, it contains a wealth of classification accuracy results for both the SV and UP datasets, making it highly valuable for analyzing the performance of various hyperspectral classification methods. Firstly, when considering the overall classification results, the SV dataset consistently outperforms the UP dataset for the same classification methods. This disparity can primarily be attributed to the larger scale of labeled training samples available in the SV dataset, which facilitates the construction of a more effective feature space. Similarly, when comparing different training ratios within the same dataset, the same trend persists. Secondly, it is evident that the 2D-CNN method, which solely focuses on spatial features and disregards spectral information, exhibits lower classification accuracy compared to other methods. This observation underscores the essentiality of leveraging spectral–spatial features for robust hyperspectral classification. Lastly, when comparing our proposed FEHN-FL framework with other methods, we consistently achieve superior classification accuracy on both the UP and SV datasets, irrespective of the training ratio employed. This demonstrates the efficacy of the focal loss in addressing class imbalance issues within hyperspectral images and refining decision boundaries during classification. Furthermore, it is noteworthy that even without considering the focal loss as the final loss function, we are still able to attain commendable classification accuracy. For instance, when utilizing the FE-HybridSN+CE framework on the SV dataset with 15% training samples, we observe exceptional accuracy levels of OA (99.97%), AA (99.95%), and kappa (99.98%). To summarize Experiment 3, we provide a set of classification performance charts in Figures 3–5, showcasing the results obtained on three distinct experimental datasets. Despite the findings from Figure 5, where our proposed method did not show an improvement in classification accuracy on the SV dataset, this result was expected. The SV dataset inherently has friendly sample feature discrimination and an ample number of labeled samples.

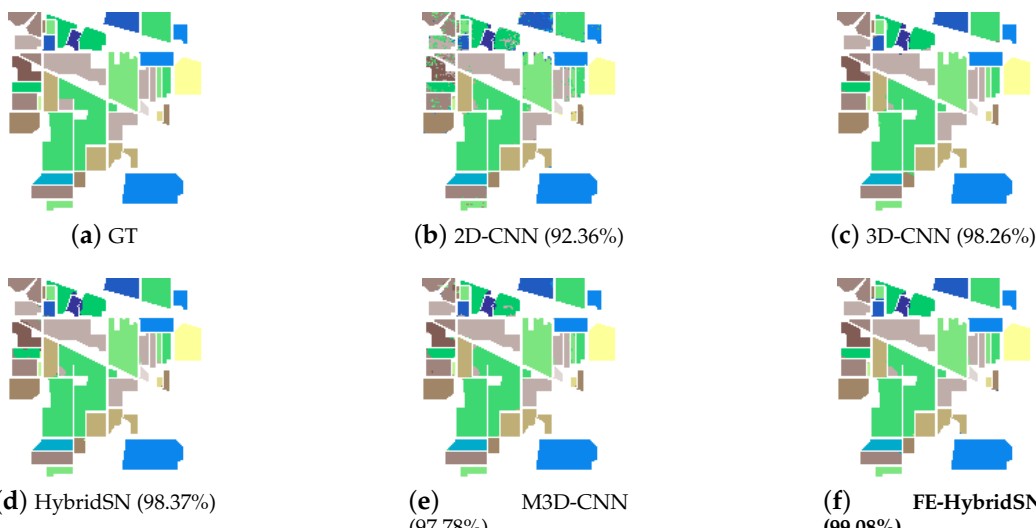

**(a)** GT     **(b)** 2D-CNN (92.36%)     **(c)** 3D-CNN (98.26%)

**(d)** HybridSN (98.37%)     **(e)** M3D-CNN (97.78%)     **(f)** **FE-HybridSN (99.08%)**

**Figure 3.** Classification maps for the IP dataset with 15% labeled training samples and $19 \times 19$ spatial window. (**a**) Ground truth. (**b**) 2D-CNN [53]. (**c**) 3D-CNN [54]. (**d**) 3D2D-HybridSN [31]. (**e**) M3D-CNN [55]. (**f**) Proposed. In the subfigure (**b**–**f**), the right brackets encompass the overall classification accuracies, and the best classfication result is highlighted in bold.

In the final experiment, we provided the learnable parameter sizes of the methods involved, which are closely related to time complexity. As shown in Tables 8 and 9, although our method's learned parameter size increased to some extent compared to other methods,

it can be observed that its training phase convergence speed is similar to, or even faster than, other methods.

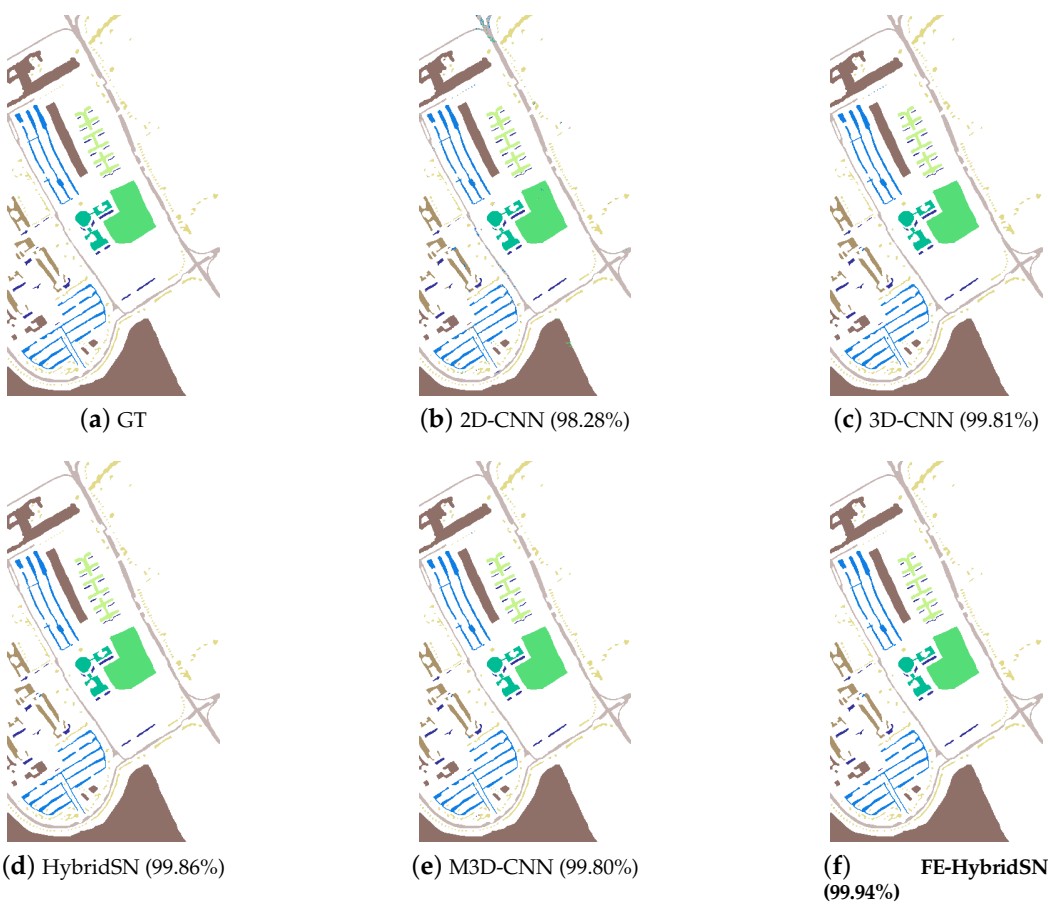

**(a)** GT    **(b)** 2D-CNN (98.28%)    **(c)** 3D-CNN (99.81%)

**(d)** HybridSN (99.86%)    **(e)** M3D-CNN (99.80%)    **(f)** **FE-HybridSN (99.94%)**

**Figure 4.** Classification maps for the UP dataset with 10% labeled training samples and $19 \times 19$ spatial window. (**a**) Ground truth. (**b**) 2D-CNN [53]. (**c**) 3D-CNN [54]. (**d**) 3D2D-HybridSN [31]. (**e**) M3D-CNN [55]. (**f**) Proposed. In the subfigure (**b**–**f**), the right brackets encompass the overall classification accuracies, and the best classification result is highlighted in bold.

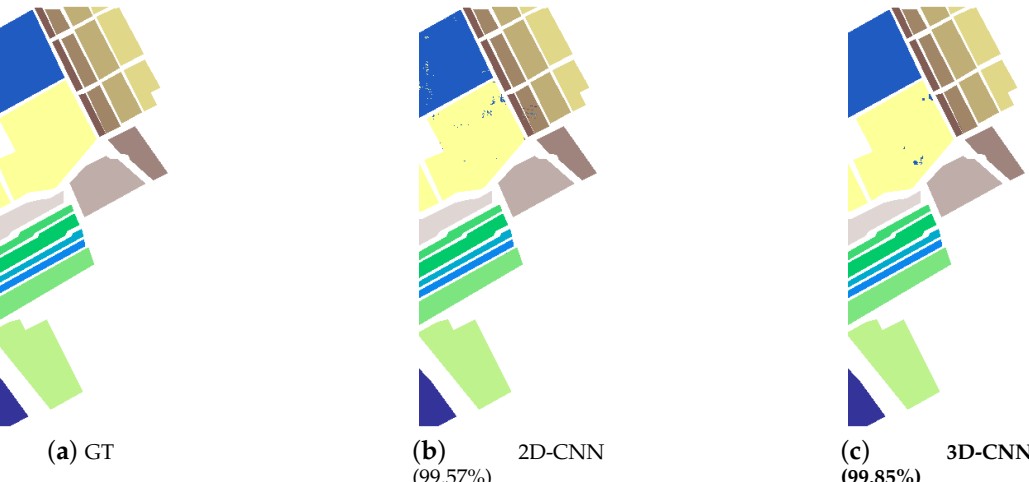

**(a)** GT    **(b)** 2D-CNN **(99.57%)**    **(c)** **3D-CNN (99.85%)**

**Figure 5.** *Cont.*

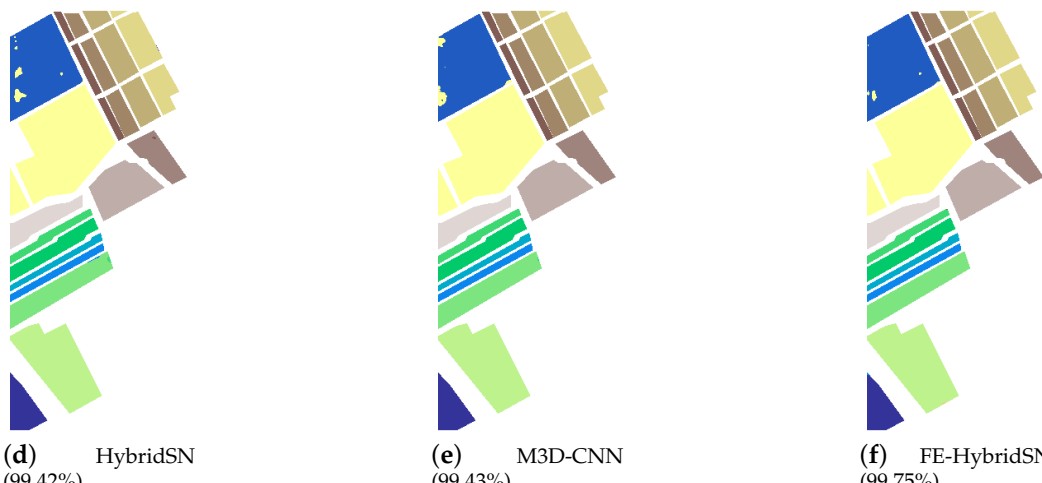

**(d)**     HybridSN
(99.42%)
**(e)**     M3D-CNN
(99.43%)
**(f)**     FE-HybridSN
(99.75%)

**Figure 5.** Classification maps for the SV dataset with 5% labeled training samples and $19 \times 19$ spatial window. (**a**) Ground truth. (**b**) 2D-CNN [53]. (**c**) 3D-CNN [54]. (**d**) 3D2D-HybridSN [31]. (**e**) M3D-CNN [55]. (**f**) Proposed. In the subfigure (**b**–**f**), the right brackets encompass the overall classification accuracies, and the best classification result is highlighted in bold.

## 5. Conclusions

The paper proposes a novel hierarchical deep neural network based on CNN for efficient extraction of spectral–spatial features from high-dimensional hyperspectral images. In the proposed framework, the hierarchical convolutional structure effectively reduces computational complexity while effectively capturing both channel and spatial texture information in the hyperspectral data. Specifically, the feature extractor FE-HybridSN consists of a three-layer 3D-CNN component dedicated to extracting spectral information across different channels, complemented by a two-layer 2D-CNN component focusing on spatial information within each channel. Compared with the state-of-the-art methods, FE-HybridSN demonstrates competitive classification performance on widely used datasets such as IP, UP, and SV. Furthermore, we introduce the focal loss as the loss function, which effectively mitigates the problem of biased classification decision boundaries caused by long-tailed distributions.

In conclusion, our research contributes to the field of hyperspectral image analysis by proposing a comprehensive framework that combines feature extraction, dimensionality reduction, and classification techniques. The experimental results demonstrate the potential of these techniques in enhancing the accuracy and efficiency of hyperspectral image classification. Although our method provides efficient performance in HSI classification, there are still several unresolved challenges that may pose future limitations. Our ongoing research will focus on the following directions to address these challenges and further improve our approach:

1. Enhancing the model design to enable adaptive adjustment of decision boundaries based on different hyperspectral datasets. This will allow our method to better accommodate the unique characteristics and variations present in different datasets.
2. Exploring and integrating advanced data augmentation techniques to tackle the issue of limited sample sizes. By generating synthetic data and applying transformational operations, we can effectively expand the training dataset and improve the model's generalization capability.
3. Investigating alternative strategies to mitigate or alleviate the impact of spatial size during the convolutional process. This includes exploring methods such as multiscale feature extraction and attention mechanisms to capture both local and global spatial information more effectively.

By addressing these areas, we aim to overcome the current limitations and further enhance the robustness, adaptability, and overall performance of our approach in hyperspectral image classification.

**Author Contributions:** Methodology, X.Y.; Validation, Y.W. and C.Y.; Formal analysis, X.W.; Writing—original draft, X.W.; Writing—review & editing, Y.S. All authors have read and agreed to the published version of the manuscript.

**Funding:** This work was funded by the National Natural Science Foundation of China under Grant [U20A2082, 41971151] and the Harbin Normal University Postgraduate Innovation Project [HSDSSCX2022-117].

**Data Availability Statement:** The related datasets in this article is open source. Our proposed framework, FEHN-FL, is implemented in Python and the referential code is available https://github.com/KeithMalarkey/HSI-FEHN-FL, accessed on 1 August 2023.

**Conflicts of Interest:** The authors declare no conflict of interest.

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
