# Peer review of "A Hybrid 3D–2D Feature Hierarchy CNN with Focal Loss for Hyperspectral Image Classification"

_remotesensing, doi:10.3390/rs15184439_

Round 1

Reviewer 1 Report

This paper presents a novel hybrid convolutional network that performs dimensionality reduction and feature extraction on high-dimensional hyperspectral image data at the method level. While reducing the computational pressure, spatial-spectral features are hierarchically extracted using the FE-HybridSN backbone network for hyperspectral classification. In addition, for the problem of class imbalance in hyperspectral data, a mitigation solution is given - introducing focal loss as a loss function, and accelerating the convergence speed of the network to a certain extent. After reading the manuscript, the following suggestions are made:

1. The positions in Figure 3 to Figure 5 are not continuous and need to be adjusted.

2. Regarding the experimental parameters, I hope to give more details.

3. In the Introduction section, the introduction of the transformer method in recent years is not comprehensive enough and has no logical level.

4. There is no reason to choose the convolutional neural network instead of the transformer as the backbone network.

5. Hope to give the computational complexity between different comparison methods.

The language in the introduction needs to be further optimized.

Reviewer 2 Report

  1. In Figure 1, the second 2D convolution is 128@K × K, which differs from the introduced 256@3 × 3 in Section 2.1 and Table 1.
  2. Batch normalization (BN) and Rectified Linear Unit (ReLU) are employed in the proposed network architecture; they can be included in Figure 1 for representation.
  3. The fully connected layers (FC) in Figure 1 are indicated as …(×3); however, the dashed green box of the FE-hybridSN does not encompass the …(×3) portion, which may lead to misunderstanding.
  4. Section 3.3.1 mentions spatial sizes in Table 3: 11×11, 15×15, 19×19, 23×23, but data for 11×11 is missing in Table 3.
  5. Could you please provide the reason for not including the IP dataset for comparison in Experiment 2 with 10% training samples?

Reviewer 3 Report

The research manuscript entitled “A Hybrid 3D-2D Feature Hierarchy CNN with Focal Loss for Hyperspectral Image Classification,” is well-written along with appropriate and enough details and descriptions. The authors introduced a combined 3D and 2D CNN framework, FE-HybridSN, for hyperspectral image classification. They integrated focal loss to handle class imbalances and tested their model on three datasets, showing competitive results. They also highlighted the significance of spatial size in accuracy and suggested future improvements. In the reviewer’s opinion, the results discussed in this manuscript are promising. However, there are few comments that needs to be addressed before it can be recommended for publication in the journal of Remote Sensing, MDPI.

1. The manuscript admits that preprocessing involves cropping the hyperspectral data into fixed-sized cubic modules, leading to a potential loss of information. This procedure could be detrimental in some remote sensing applications where edge and boundary information is crucial. A more detailed exploration or analysis of the impact of this information loss should be included.

2. While the authors mentioned that FE-HybridSN effectively captures more spectral-spatial information while reducing computational complexity compared to standalone 3D-CNNs, there is, in fact, no mention of a direct comparison between the FE-HybridSN and plain 3D-CNNs. It would be beneficial if the authors provide a side-by-side comparison to justify this claim.

3. The choice of using focal loss is interesting, especially in mitigating class imbalance. However, it would be beneficial to know more about how focal loss parameters were chosen and if other advanced loss functions were considered or compared against.

4. Although experiments were conducted on three datasets (IP, UP, SV), how does FE-HybridSN perform on other hyperspectral datasets, especially those with different characteristics or from different domains?

5. Since the manuscript aims at reducing computational complexity, it would be useful to know the actual computational times and the hardware used, allowing readers to gauge the true benefits in a real-world context.

6. The abstract and conclusion of this manuscript, while comprehensive, can benefit from a more concise presentation, removing redundancy and focusing on the major contributions and findings.

7.  Additional details on how experiments were conducted, such as data splits for training, validation, and testing, or hyperparameters used, would provide more clarity.

8. While the conclusion gives a roadmap for future research directions, it would be interesting to see preliminary results if the authors have already started on any of these paths.

9. Ensure that all state-of-the-art methods, datasets, and foundational concepts mentioned are adequately cited.

Ensure that the manuscript is proofread to eliminate any grammatical or syntactical errors, which can detract from the overall clarity and professionalism of the work.

Round 2

Reviewer 2 Report

The authors provided the detailed responses and my concerns have been addressed.  

Reviewer 3 Report

The authors of the paper titled “A Hybrid 3D-2D Feature Hierarchy CNN with Focal Loss for Hyperspectral Image Classification” have effectively responded to my feedback and significantly enhanced the quality of the manuscript. I believe it now meets the necessary criteria for publication. The current manuscript can be accepted for publication in the journal of Remote Sensing, MDPI.

In the reviewer's opinion, the quality of English can be acceptable; but minor editing is needed.